# DETECTING MEMORIZATION IN ReLU NETWORKS

## ABSTRACT

We propose a new notion of 'non-linearity' of a network layer with respect to an input batch that is based on its proximity to a linear system, which is reflected in the non-negative rank of the activation matrix. We measure this non-linearity by applying non-negative factorization to the activation matrix. Considering batches of similar samples, we find that high non-linearity in deep layers is indicative of memorization. Furthermore, by applying our approach layer-by-layer, we find that the mechanism for memorization consists of distinct phases. We perform experiments on fully-connected and convolutional neural networks trained on several image and audio datasets. Our results demonstrate that as an indicator for memorization, our technique can be used to perform early stopping.

## 1 INTRODUCTION

A fundamental challenge in machine learning is balancing the bias-variance tradeoff, where overly simple learning models underfit the data (suboptimal performance on the training data) and overly complex models are expected to overfit or *memorize* the data (perfect training set performance, but suboptimal test set performance). The latter direction of this tradeoff has come into question with the observation that deep neural networks do not memorize their training data despite having sufficient capacity to do so (Zhang et al., 2016), the explanation of which is a matter of much interest.

Due to their convenient gradient properties and excellent performance in practice, rectified-linear units (ReLU) have been widely adopted and are now ubiquitous in the field of deep learning. In addition, the relative simplicity of this function $(\max(\cdot, 0))$ makes the analysis of ReLU networks more straight-forward than networks with other activation functions.

We propose a new notion of 'non-linearity' of a ReLU layer with respect to an input batch. We show that networks that generalize well have deep layers that are approximately linear with respect to batches of similar inputs. In contrast, networks that memorize their training data are highly non-linear with respect to similar inputs, even in deep layers.

Our method is based on the fact that the main source of non-linearity in ReLU networks is the threshold at zero. This thresholding determines the *support* of the resulting activation matrix, which plays an important role in the analysis of non-negative matrices. As we discuss in Section 3, the *non-negative rank* of a matrix is constrained by the shape of the support, and is therefore indicative of the degree of non-linearity in a ReLU activation matrix with respect to the input.

Although computing the non-negative rank is NP-hard (Vavasis, 2009), we can restrict it with approximate *non-negative matrix factorization* (NMF) (Lee & Seung, 1999). Consequently, we propose to estimate the 'non-linearity' of a ReLU layer with respect to an input batch by performing NMF on a grid over the approximation rank $k$, and measuring the impact on network performance.

This procedure can be seen as measuring the robustness of a neural network to increasing compression of its activations. We therefore compare our NMF-based approach to two additional dimensionality reduction techniques, namely principal component analysis (PCA) and random ablations.

We informally define memorization as the implicit learning of a rule that associates a specific sample (i.e., with index $i$) to a particular label (e.g., with index $j$). Such a rule does not benefit the network in terms of improving its performance on new data.

We show that our NMF-based approach is extremely sensitive to memorization in neural networks. We report results for a variety of neural network architectures trained on several image and audio

datasets. We conduct a layer-by-layer analysis and our results reveal interesting details on the internal mechanism of memorization in neural networks. Finally, as an indicator for memorization, we use our proposed measure to perform early stopping.

## 2   RELATED WORK

The study of factors involved in the bias-variance tradeoff in learning models goes back several decades. Classical results in statistical learning consider properties of learning models such as the VC-dimension (Vapnik, 1998) and Rademacher complexity (Bartlett & Mendelson, 2002). These properties give generalization bounds in terms of the capacity model to (over)fit data. When considering the vast capacity of deep neural networks, such bounds become irrelevant and fail to explain their ability to generalize well in practice (Zhang et al., 2016; Bartlett et al., 2017).

More direct analyses have been done with respect to a specific setting of model parameters. For instance, Bartlett (1998) showed that the number of weights in a network is less important compared to their scalar value (e.g. $\ell_2$-norm), and more recently Bartlett et al. (2017) presented a bound for deep neural networks based on the product of spectral norms of the network's weight matrices. Achille & Soatto (2017) showed that memorizing networks contain more information in their weights.

Methods to explain generalization have been proposed that examine a network's robustness to perturbations (Hochreiter & Schmidhuber, 1997; Chaudhari et al., 2016; Keskar et al., 2016; Neyshabur et al., 2017; Li et al., 2017). These methods propose the notion of *flatness of minima* on the loss surface, assuming that perturbing the parameters without dramatically changing performance is an indicator of the generalization of a network.

However, any reversible transformation, such as simple scaling, can arbitrarily manipulate the local flatness without affecting generalization (Dinh et al., 2017). The procedure we propose can be viewed as applying perturbations, albeit to *activations* and not parameters, and must address this concern. The perturbations we apply to activations account for magnitude, since they depend on a change of rank or non-negative rank of the activation matrix, a property which is robust to rescaling and similar reversible transformations.

In contrast to the methods described thus far, which deal exclusively with the parameters of the model, methods have been developed that account for the role of the data distribution. Liang et al. (2017) proposed to use the Fisher-Rao norm, which uses the geometry of the data distribution to weigh the contribution of different model parameters. The empirical studies of Morcos et al. (2018) and Novak et al. (2018) explore robustness to specific types of noise. The former uses Gaussian noise and masking noise injected into hidden activations, while the latter interpolates between input samples to study network behavior on and off the data manifold. In both cases, robustness to noise proved a reliable indicator for good generalization. Additionally, Arora et al. (2018) derive generalization bounds in terms of robustness to noise.

Our experimental setup is reminiscent of Morcos et al. (2018) in that both methods apply a form of compression to hidden activations and test for robustness to this type of noise. Specifically, they set random axis-aligned directions in feature space to zero which can be viewed as a crude form of dimensionality reduction, i.e., by simply removing canonical dimensions. In our experiments we refer to this method as random ablations. Our results show that robustness to NMF compression is much more correlated with low memorization/high generalization than robustness to random ablations. Arpit et al. (2017) have also studied various empirical aspect of memorization.

As a dimensionality reduction technique, NMF has gained popularity due to its producing meaningful factorizations that lend themselves to qualitative interpretation across various domains such as document clustering (Xu et al., 2003), audio source separation (Grais & Erdogan, 2011), and face recognition (Guillamet & Vitria, 2002). In the context of deep convolutional neural networks, Collins et al. (2018) applied NMF to the activations of an image classifier and showed that the result gives a decomposition into semantic parts, which benefits from the transformation invariance learned by the neural network.

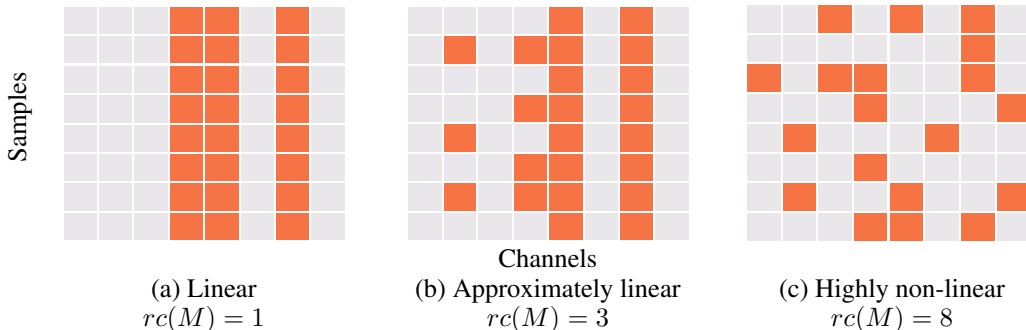

(a) Linear
$rc(M) = 1$

(b) Approximately linear
$rc(M) = 3$

(c) Highly non-linear
$rc(M) = 8$

Figure 1: **The support of the activation matrix** is determined by ReLU threshold. (a) When all the rows of the support are identical, there is a sub-weight-matrix such that the layer is fully linear with respect to the input batch. (b, c) As the support becomes more complex, which we characterize by the increase in its rectangle cover number, the layer becomes more non-linear.

## 3 METHOD

Consider a ReLU layer of a neural network, parameterized by a weight matrix $\boldsymbol{W} \in \mathbb{R}^{m \times q}$. For a batch of $n$ inputs $\boldsymbol{X} \in \mathbb{R}^{n \times m}$, we compute the layer activation matrix $\boldsymbol{A}$ as follows:

$$\boldsymbol{A} = \max\left(\boldsymbol{X}\boldsymbol{W}, 0\right) \in \mathbb{R}_+^{n \times q}, \tag{1}$$

where $\mathbb{R}_+$ are the non-negative reals. We omit the bias term for notational convenience.

### 3.1 ReLU-INDUCED SUPPORT

The processing of a single input $\boldsymbol{x}$ by a ReLU network is equivalent to sampling a sub-network that is linear with respect to the sample (Wang et al., 2016). This could be accomplished by simply setting to zero the columns of each $\boldsymbol{W}$ whose dot product with the input is negative (and would thus be set to zero by ReLU), and then removing the thresholding.[1]

Extending this notion to a batch of several input samples to a ReLU layer, suppose the samples are sufficiently close to each other such that they all share the same ReLU mask $\boldsymbol{m} \in \{0, 1\}^q$. In this case, we may say that *the layer is linear with respect to its input batch*. This is because, for the entire batch, instead of using ReLU, we could zero out a subset of columns and obtain a linear system, i.e., $\boldsymbol{A} = \boldsymbol{X}\boldsymbol{W}\mathrm{diag}(\boldsymbol{m})$.

For an activation matrix $\boldsymbol{A}$ (Equation 1), we consider the support $\boldsymbol{M} = \mathrm{supp}(\boldsymbol{A})$, which we describe as a binary 0/1 matrix where $\boldsymbol{M}_{i,j} = 1$ where $\boldsymbol{A}_{i,j} > 0$. Because $\boldsymbol{A}$ is a ReLU activation matrix, the structure of $\boldsymbol{M}$ is mainly determined by the thresholding at zero.[2] Because thresholding is the main source of non-linearity in a ReLU network, the support takes on a special meaning in this case.

### 3.2 RECTANGLE COVER NUMBER AND NON-NEGATIVE RANK

We want to characterize how close to being linear a layer is with respect to its input $\boldsymbol{X}$ by examining the support of the resulting activations $\boldsymbol{M}$. If all the rows of $\boldsymbol{M}$ are identical to a unique vector $\boldsymbol{m}$, we can say the layer is completely linear with respect to $\boldsymbol{X}$. In general, the 'simpler' the support $\boldsymbol{M}$, the closer to linearity the layer.

One measure that captures this idea is the *rectangle cover number* of a matrix, $rc(\boldsymbol{M})$, an important quantity in the study of communication complexity (Klauck, 2003). Also known as the *Boolean rank*, $rc(\boldsymbol{M})$ is the smallest number $r$ for which there exist binary matrices $\boldsymbol{U}_B \in \{0, 1\}^{n \times r}, \boldsymbol{V}_B \in$

---

[1]This is a similar intuition to that of viewing dropout as an approximation to a model ensemble, where the dropout mask is seen to sample a sub-network (Srivastava et al., 2014).

[2]The probability of an activation value being exactly zero prior to thresholding is negligible.

$\{0, 1\}^{r \times q}$ such that their *Boolean* matrix multiplication satisfies $\boldsymbol{M} = \boldsymbol{U}_B \boldsymbol{V}_B$. As a complexity measure for ReLU activations, $rc(\boldsymbol{M}) = 1$ means the layer is linear with respect to its input, and higher values $rc(\boldsymbol{M})$ imply increasing non-linearity. This is visualized in Figure 1.

Intuitively, imagine having to fit a layer with 'ReLU switches', each of which controls a subset of weight matrix columns. In the linear case, one switch would suffice to describe the data. In the most non-linear case, we would require a switch for every column, which is also the maximal value of $rc(\boldsymbol{M})$.

Because computing the rectangle cover number $rc(\boldsymbol{M})$ is complex, several approximations and bounds to it have been studied (Fiorini et al., 2013). For the support of a non-negative matrix, a well-known upper-bound is:

$$rc(\text{supp}(\boldsymbol{A})) \leq rank_+(\boldsymbol{A}), \tag{2}$$

where $rank_+(A)$ is the *non-negative rank* of $\boldsymbol{A}$ (Gillis & Glineur, 2012) that is defined as the smallest number $k$ such that there exist non-negative matrices $\boldsymbol{U}_+ \in \mathbb{R}_+^{n \times k}$, $\boldsymbol{V}_+ \in \mathbb{R}_+^{k \times q}$ such that $A = U_+ V_+$. Similar to the rectangle cover number, the non-negative rank is hard-constrained by the combinatorial arrangement of $\text{supp}(\boldsymbol{A})$, but additionally accounts for the actual value in the non-zero entries of $\boldsymbol{A}$.

While computing $rank_+(\boldsymbol{A})$ is not easier than computing $rc(\text{supp}(\boldsymbol{A}))$, we can *restrict it* by performing approximate non-negative matrix factorization (NMF).

## 3.3 Non-negative matrix factorization

For a given non-negative rank constraint $k$, NMF solves for:

$$\underset{\boldsymbol{U}_+, \boldsymbol{V}_+}{\arg\min} \|\boldsymbol{A} - \boldsymbol{U}_+ \boldsymbol{V}_+\|_2^2, \tag{3}$$

with $\boldsymbol{U}_+, \boldsymbol{V}_+$ as defined above. The result $\boldsymbol{U}_+ \boldsymbol{V}_+ = \tilde{\boldsymbol{A}}_k \approx \boldsymbol{A}$ is the closest matrix to $\boldsymbol{A}$ under the Frobenius norm that has $rank_+$ at most $k$.

Consequently, we propose to estimate the 'linearity' of a ReLU layer with respect to a batch of similar inputs by performing NMF on a grid over the non-negative rank $k$, and measuring the impact on network performance by observing the change in the prediction (output layer) as we change $k$. This procedure also addresses the fact that in practice network activations tend to be noisy, whereas $\text{supp}(\boldsymbol{A})$ is not robust to noise, i.e., $\boldsymbol{A}_{i,j} = \epsilon > 0 \rightarrow \boldsymbol{M}_{i,j} = 1$ even for very small $\epsilon$.

Concretely, if we let $\boldsymbol{A}_i$ be the activation matrix at layer $i$, during the forward pass, we replace the feature activations of one or several layers with their rank $k$ NMF approximations:

$$\boldsymbol{A}_{i+1} = \max\left(\tilde{\boldsymbol{A}}_k W_{i+1}, 0\right) \tag{4}$$

For convolutional networks, we first reshape the tensor of feature maps from $n \times q \times h \times w$ to $(n \cdot h \cdot w) \times q$, i.e., we flatten the batch $(n)$ and spatial dimensions $(h, w)$ to form a matrix with $q$ columns, where $q$ is the number of channels in that layer. We then inversely reshape the approximated features to continue forward propagation through the network.

## 3.4 Single class batches

We now characterize the input batch, with respect to which we would like to measure layer linearity. Informally, the goal of training is to cluster together input samples that have similar (or identical) output labels,while separating them from samples of other labels. In the context of classification then, we expect therefore that from a certain depth and onward, samples of the same class will have similar activations, and thus a simpler support. In other words, while a network may exploit flexible non-linear structure to separate *different classes*, we expect that with respect to a *single class*, deep layers are approximately linear.

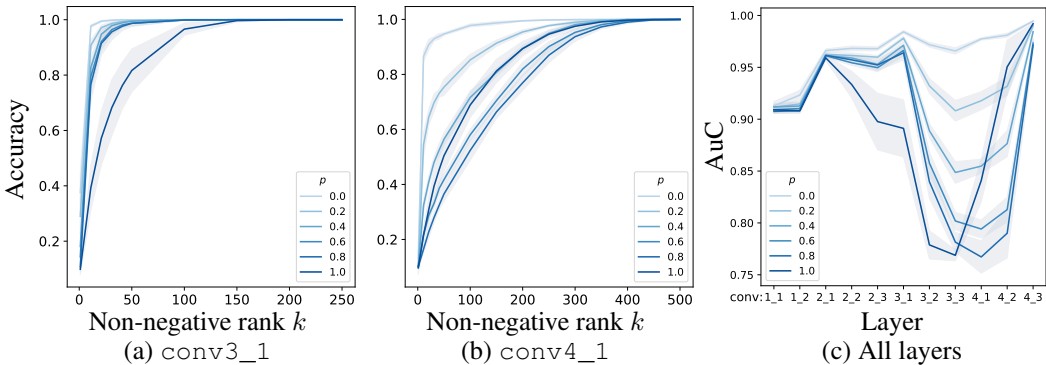

Figure 2: **Memorization mechanism** of CNNs trained on CIFAR-10, with increasing level of label randomization $p$ (i.e., $p = 0$ is the unmodified dataset). We analyze each layer by applying NMF compression to its activation matrix with increasing rank $k$, while observing the impact on classification performance. In (a) and (b) we show the $k$ vs. accuracy curves at an layers of different depth. We can immediately see that in deep layers, networks with high memorization are significantly less robust to NMF compression, indicating higher degrees of non-linearity. Furthermore, networks trained on fully randomized labels ($p = 1$) behave differently than networks with partial or no randomization. By summarizing each curve in (a) and (b) by its area under the curve (AuC), we show in (c) a birds-eye view over all layers. All networks with $p < 1$ pass through distinct phases consisting of a *feature extraction* phase until conv3_1, followed by a *memorization* phase until conv4_2, followed by a final *clustering* phase. Interestingly, the case $p = 1$ shifts the process into earlier layers, explaining why layer-by-layer it appears as an outlier.

When single-class batches are *not* approximately linear even in deep layers, i.e., activations are not clustered within a few linear regions, we empirically show in the next section that this behavior is indicative of memorization.

## 4 EXPERIMENTS

### 4.1 FEATURE COMPRESSION AND MEMORIZATION

We start by studying networks that have been forced into different levels of memorization due to label randomization applied to their training set (Zhang et al., 2016). The level of induced memorization is controlled by setting a probability $p$ for a training label to be randomized, i.e., $p = 0$ is the unmodified dataset and $p = 1$ gives fully random labels. Note that the capacity of these networks is sufficiently large such that the training accuracy is 1 in all cases, regardless of the value of $p$.

As such, we use batches of training data and observe how accuracy drop from 1 to constant prediction as we increase the level of compression. In all experiments, sampling single-class batches is done with respect to the label used for training (i.e., the random label if $p > 0$). We sample batches stochastically (up to the label), have found all methods discussed below to be robust to the batch size (e.g., 20-100). In all our experiments we set the batch size to 50.

We perform experimental evaluations on several image datasets, namely CIFAR-10 (Krizhevsky & Hinton, 2009), Fashion-MNIST (Xiao et al., 2017), SVHN(Netzer et al., 2011), and ImageNet (Russakovsky et al., 2015), as well as on the Urban Sounds audio classification dataset (Salamon et al., 2014). We use a fully-connected network for Fashion-MNIST and various CNN architectures for the others, which we describe in more detail the appendix.

### 4.1.1 LAYER BY LAYER ANALYSIS

We start by analyzing the layers of an 11-layer CNN trained on CIFAR-10. We sampled 10 batches (one batch per class) of 50 images, and compressed the activation matrix at each layer individually down to various values of the non-negative rank. We then measured classification accuracy of the

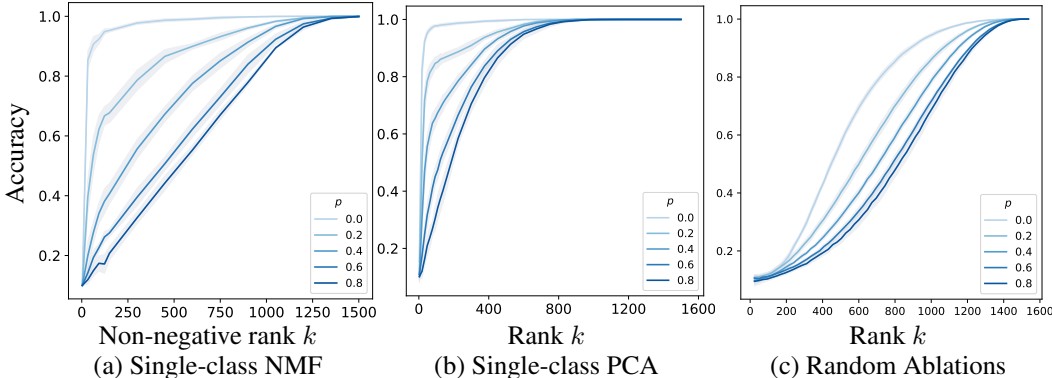

Figure 3: **Detecting memorization via compression**. We demonstrate this on networks trained with different levels of label randomization ($p$), and hence of memorization. (a) Due to its sensitivity to the non-linearity of ReLU activations, NMF successfully captures the level of memorization present in neural networks. (b) PCA compression is able to regain sufficient variance for good accuracy even with small values of $k$, which renders it less effective for detecting memorization. (c) Though not taking into account batch statistics, random ablations can distinguish between different levels of memorization, albeit less significantly. Compression was applied to the final three (convolutional) layers of CNNs trained on CIFAR-10.

prediction. In this analysis we report average results for 60 neural networks, ten networks (with different random initializations) trained per randomization level $p$.

In Figure 2 (a) and (b) we show $k$ vs. accuracy curves of networks trained with increasing levels of label randomization, at an early layer (`conv2_1`) and a deep layer (`conv4_1`) respectively. We can immediately see that networks trained on fully randomized labels ($p = 1$) behave differently than networks with partial or no randomization. Furthermore, note that in deep layers, memorizing networks are significantly less robust to NMF compression, i.e., their activations posses a high non-negative rank, which indicates high non-linearity with respect to the input, as discussed in Section 3.2. We can characterize each curve in (a) and (b) with a single number, its area under the curve (AuC). This allows us in Figure 2 (c) to generate a single figure for all layers. Networks with $p < 1$ display a similar feed-forward trend up until layer `conv3_1`. Since these networks differ from each other in no way other than the level of label randomization on the training data, we hypothesize this to be a generic *feature extraction* phase common to all of them. In the next phase, until `conv4_2`, we see a big difference between networks, such that more memorization (higher $p$) is correlated with lower AuC, i.e., higher non-negative rank and hence non-linearity of those layers with respect to single-class batches. We therefore localize memorization to these layers. Lastly, the phase only of `conv4_3` is where samples of the same class are clustered together, right before the final 10-dimensional classification layer (which is not shown). This final phase is in accordance with the premise that regardless of randomization level $p$, all of these networks achieve perfect training set accuracy. Interestingly, setting $p = 1$ shifts the process to earlier layers, explaining why layer-by-layer this case appears as an outlier.

### 4.1.2 COMPRESSION TECHNIQUES

We compare different compression techniques with regards to detecting memorization. Given our choice of solving NMF under the Frobenius norm (Equation 3), a natural method to compare against is principal component analysis (PCA), which optimizes under the same norm but without the non-negativity constraint on its factors. We also consider random ablations, i.e., setting a random subset of columns in the activation matrix to zero, since this technique has been used previously to detect memorization (Morcos et al., 2018).

Rather than choosing a single layer, we sequentially apply compression to several layers. We target the final convolutional blocks of our CNNs, all of which contain three layers, each of which consists of 512 channels. In fully-connected networks, we applied compression to all layers.

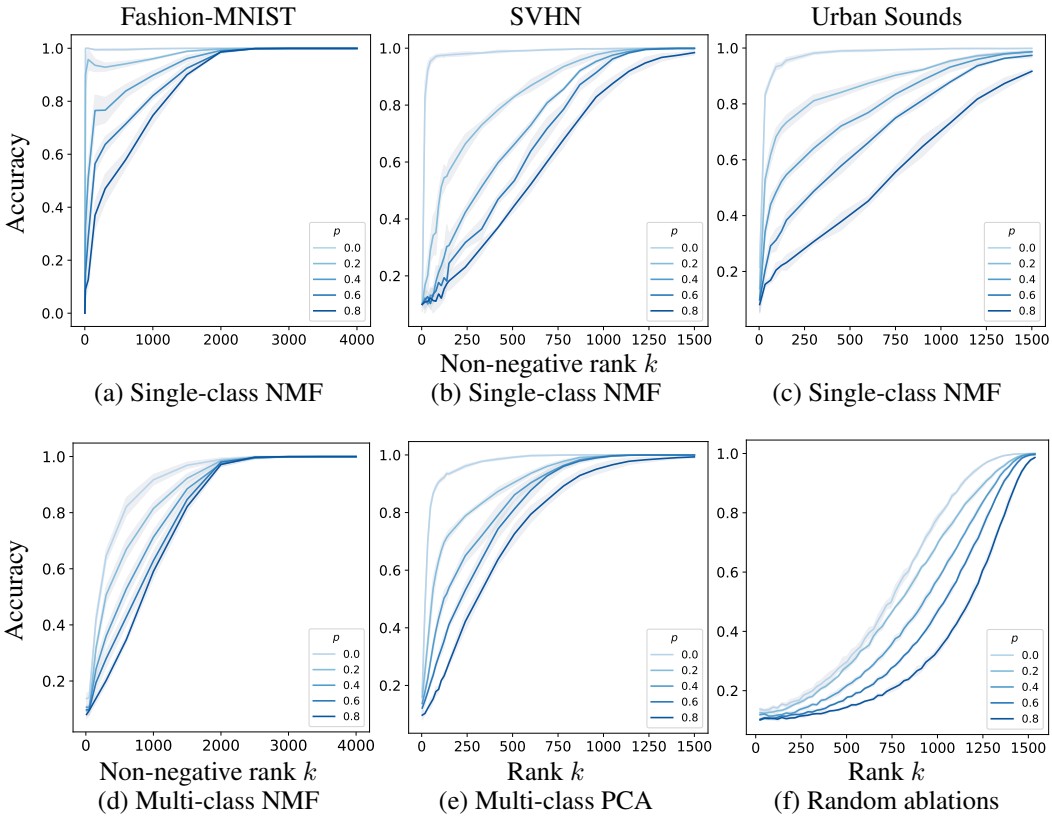

Figure 4: **Memorization across various datasets and network architectures**. We show that NMF-based compression is sensitive to memorization in diverse settings. Each column shows results for a specific dataset and network architecture. (a, b, d, e) We show that network layers are considerably more linear with respect to single-class batched than with respect to multi-class batches. (b,c,e,f) PCA and random ablations show less sensitivity to memorization compared with NMF.

In Figure 3 we give results for the CIFAR-10 dataset, which confirm that NMF compression is indeed more sensitive to memorization, due to the properties of the non-negative rank discussed in Section 3.2. PCA, which is less constrained, is more 'efficient' at compressing the activations, but is in turn less discriminative with respect to the level of memorization. Finally, we confirm that robustness to random ablations correlates with less memorization, however less so than NMF. It should be noted that NMF does show more variance than the other two methods, and incurs a higher computational cost, as discussed in section 6.6 in the appendix.

In Figure 4 we show additional results for single-class NMF on three additional datasets and network architectures (described in the appendix), including a fully-connected network for Fashion-MNIST. The results in (d) and (e), of applying PCA and NMF to multi-class batches, show that such batches produce activations with higher rank or non-negative rank compared to single-class batches. This is a result of the network trying to separate samples of different labels.

## 4.2 FEATURE COMPRESSION AND GENERALIZATION

We have shown results for networks forced into memorization due to label randomization. In this section we show our technique is useful for predicting good generalization in a more realistic setting, without artificial noise.

In addition to the experiments below, we refer the reader to Section 6.3 in the appendix, where predict *per-class* generalization of a pre-trained VGG-19 network on a ImageNet classes.

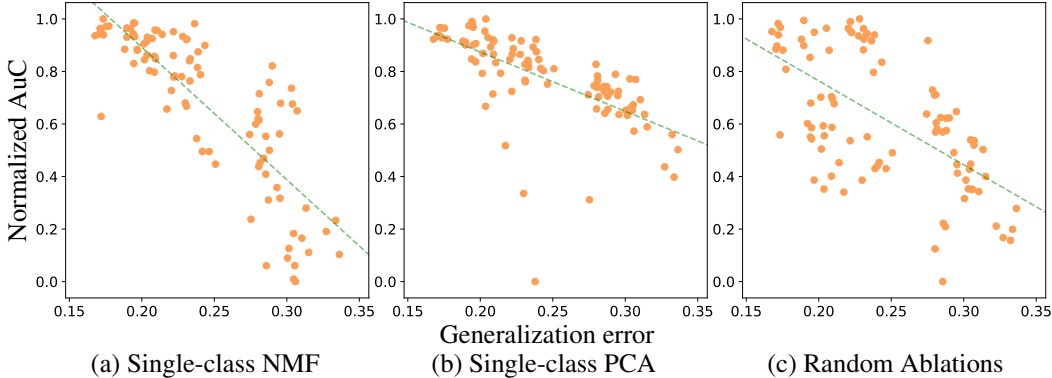

Figure 5: **Detecting generalization via compression**. While all three methods show correlation with generalization error, NMF is most correlated with a Pearson correlation of -0.82, followed by PCA with -.064 and random ablation with -0.61.

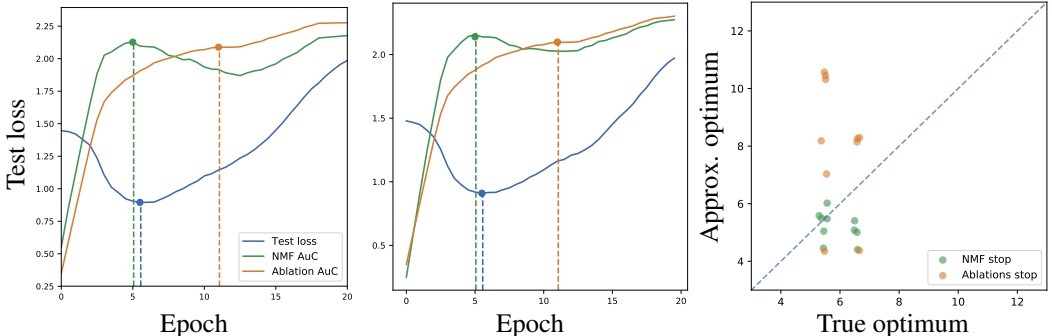

Figure 6: **Early stopping** for CNN training on CIFAR-10. (a, b) The test loss is (in blue) starts to increase after about the 5th epochs, indicating the start of overfitting. Using our proposed single-class NMF approach, we can detect the test loss turning point. We show the area under the curve (AuC) for the single-class NMF approach (in green) for the accuracy measures as discussed in Section 4.1.2. Similarly, we show the AuC when performing random ablations (in orange). (c) The NMF AuC curve and test loss curve consistently have near extrema, as seen over several runs.

### 4.2.1 COMPRESSION TECHNIQUES

We trained 96 CNN classifiers on CIFAR-10, over a grid of hyper-parameter values for the batch size, weight decay and optimization algorithm, SGD vs. ADAM (Kingma & Ba, 2015). Following the same procedure as above, for each of the three methods, NMF, PCA, or random ablations, we computed the $k$ vs. accuracy curves for each network, targeting its final convolutional block. In Figure 5 we compare the area under the curve (AuC) of each curve with the average generalization error on the test set.

While all three methods show correlation with generalization error, NMF is most correlated with a Pearson correlation of -0.82, followed by PCA with -.064 and random ablation with -0.61.

### 4.2.2 EARLY STOPPING

We test whether our method can detect memorization during training. Doing so would allow us to perform early stopping, i.e., stop training as memorization begins to decrease generalization.

We trained CNNs on CIFAR-10 with the original labels. Each network was trained for 10K batches with a batch size of 100. We recorded the test set error every 250 batches, and tested the non-linearity of the deepest three convolutional layers using our NMF-based approach with a coarse grid on $k$. As before, we compute the area under each $k$ vs. accuracy curve as in Figures 3 (c). Finally, we also computed the area under the curve produced by random ablations.

Results of two instances are shown in Figure 6 (a) and (b). In these figures we compare the test loss against our single-class NMF method and random ablations. We smooth the plots using a radius of two epochs to reduce the noise. The matching-color dashed lines mark the local minima of the test loss in as well as the location of the first local maxima of the NMF and random ablation AuC curves after smoothing has been applied. We notice that the test loss minima align almost precisely with the maximum NMF AuC. We further confirm this behavior in Figure 6 (c), where we compare the stopping times of NMF and the random ablations method against the best test loss over 10 different runs.

## 5   Conclusion

We have introduced a notion of a ReLU layer's non-linearity with respect to an input batch, which is based on its proximity to a linear system. We measure this property indirectly via NMF applied to deep activations of single-class batches. While more analysis is required before definite guarantees could be given, we find that our approach is successful in detecting memorization and generalization across a variety of neural network architectures and datasets.

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

## 6 APPENDIX

### 6.1 NEURAL NETWORK ARCHITECTURES

| CIFAR-10 | | | | | Urban Sounds | | | | |
|----------|---------|--------|---------|--------|--------------|---------|--------|---------|--------|
| Type | Out dim. | Kernel | Padding | Stride | Type | Out dim. | Kernel | Padding | Stride |
| Conv$_+$ | 64 | 3 | 1 | 1 | Conv$_+$ | 64 | 3 | 1 | 1 |
| Conv$_+$ | 64 | 3 | 1 | 1 | MaxPool | - | 2 | - | 1 |
| Conv$_+$ | 128 | 3 | 1 | 2 | Conv$_+$ | 128 | 3 | 1 | 1 |
| Conv$_+$ | 128 | 3 | 1 | 1 | Conv$_+$ | 128 | 3 | 1 | 1 |
| Conv$_+$ | 128 | 3 | 1 | 1 | MaxPool | - | 2 | - | 1 |
| Conv$_+$ | 256 | 3 | 1 | 2 | Conv$_+$ | 256 | 3 | 1 | 1 |
| Conv$_+$ | 256 | 3 | 1 | 1 | Conv$_+$ | 256 | 3 | 1 | 1 |
| Conv$_+$ | 256 | 3 | 1 | 1 | MaxPool | - | 2 | - | 1 |
| Conv$_+$ | 512 | 3 | 1 | 2 | Conv$_+$ | 512 | 3 | 1 | 1 |
| Conv$_+$ | 512 | 3 | 1 | 1 | Conv$_+$ | 512 | 3 | 1 | 1 |
| Conv$_+$ | 512 | 3 | 1 | 1 | MaxPool | - | 2 | - | 1 |
| Linear | 10 | - | - | - | Linear$_+$ | 4096 | - | - | - |
| | | | | | Linear$_+$ | 4096 | - | - | - |
| | | | | | Linear | 10 | - | - | - |

| SVHN | | | | | Fashion-MNIST | | | | |
|------|---------|--------|---------|--------|---------------|---------|--------|---------|--------|
| Type | Out dim. | Kernel | Padding | Stride | Type | Out dim. | Kernel | Padding | Stride |
| Conv$_+$ | 64 | 3 | 1 | 1 | Linear$_+$ | 128 | - | - | - |
| Conv$_+$ | 64 | 3 | 1 | 1 | Linear$_+$ | 512 | - | - | - |
| Conv$_+$ | 128 | 3 | 1 | 2 | Linear$_+$ | 2048 | - | - | - |
| Conv$_+$ | 128 | 3 | 1 | 1 | Linear$_+$ | 2048 | - | - | - |
| Conv$_+$ | 256 | 3 | 1 | 2 | Linear | 10 | - | - | - |
| Conv$_+$ | 256 | 3 | 1 | 1 | | | | | |
| Conv$_+$ | 512 | 3 | 1 | 2 | | | | | |
| Conv$_+$ | 512 | 3 | 1 | 1 | | | | | |
| Linear | 10 | - | - | - | | | | | |

Table 1: Neural architecture used for each dataset in Section 4.

The exact architectures we used for each dataset are given in Table 1. We denote a linear or convolutional layer followed by a ReLU as Linear$_+$ and Conv$_+$, respectively.

### 6.2 ABLATING NMF AND PCA DIRECTIONS

It is interesting to study the impact of *ablating* the activation in the directions found by NMF and PCA by forward propagating the residual, i.e.,

$$\boldsymbol{A}_{i+1} = \max\left((\boldsymbol{A}_k - \tilde{\boldsymbol{A}}_k)W_{i+1}, 0\right) \qquad (5)$$

This is interesting because in the case of PCA, for instance, the top $k$ directions are those that capture most of the variance in the activation matrix, and presumably the $k$ directions found by NMF are of similar importance. This is not true for the random ablations, where the ablated directions are of no special importance.

In Figure 7 we see that networks with no induced memorization that are *most vulnerable* to ablation of NMF and PCA direction. In other words, while non-memorizing networks are more robust to *random* ablations, they are not robust to ablations of specific important directions. This is in contrast to the interpretation of Morcos et al. (2018) that non-memorizing networks are more robust to ablations of single directions.

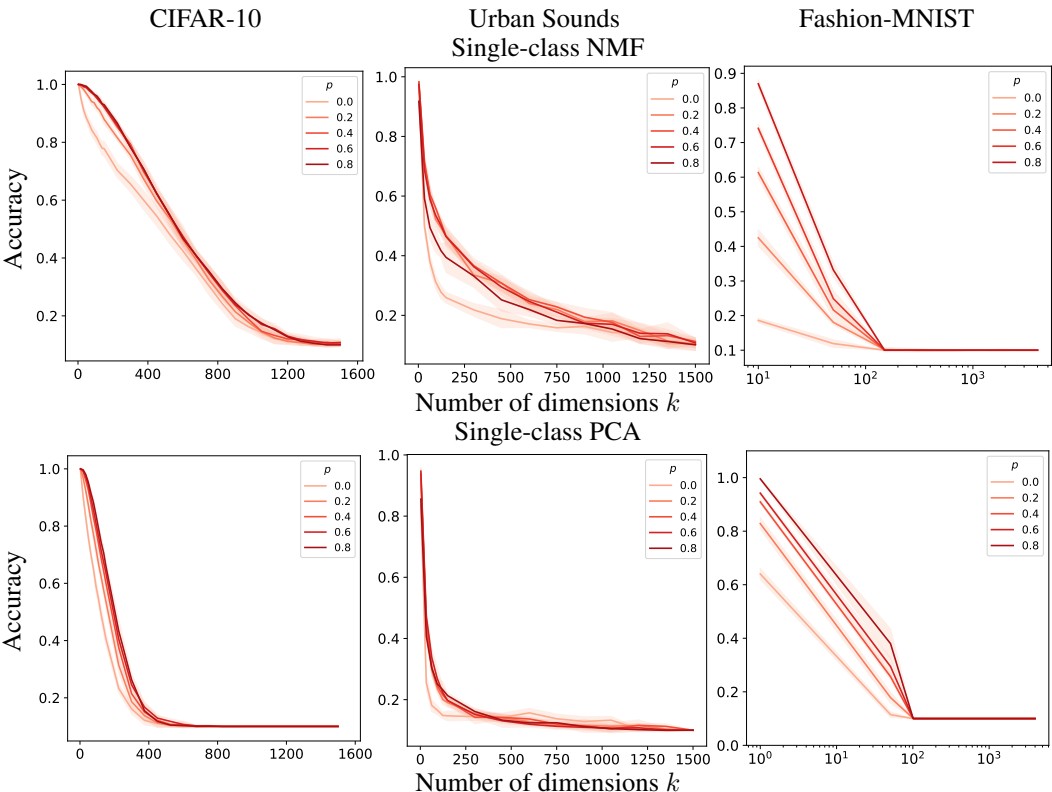

Figure 7: **NMF and PCA directions are more important** for networks with induced-memorization. Each column shows results for a specific dataset and network architecture. Compared to random ablation as in Figures 3 and 4, where networks with induced-memorization are most robust, in all cases we see that ablation in NMF and PCA directions hurts their performance more, compared to memorizing networks.

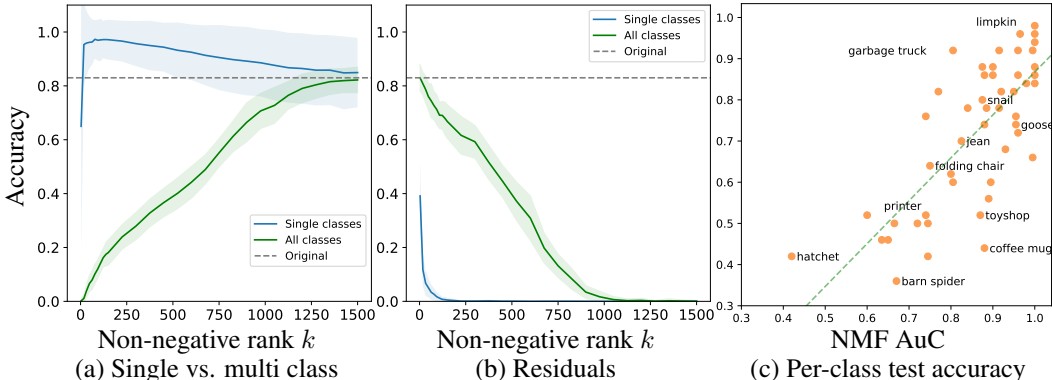

Figure 8: **NMF compression on VGG-19**. (a) Deep VGG layers are highly linear with respect to single-class batches, as indicated by high accuracy for small dimensions of $k$. Compression can have a denoising effect and improve upon the baseline accuracy of the batch (dashed line). (b) Removing NMF directions causes a dramatic drop in accuracy, more so on single-class batches. (c) Per-class test set accuracy is significantly correlated with the area under the $k$ vs. accuracy curve (NMF AuC).

## 6.3 NMF ON VGG-19

The VGG-19 model (Simonyan & Zisserman, 2014), trained on ImageNet (Russakovsky et al., 2015), is known for its good generalization ability, as evident by its widespread use as a general feature extractor. We use a pre-trained model here as an example of a well-generalizing network and analyze it with our method.

We apply NMF compression to the three deepest convolutional layer, on activations of both single-class batches and multi-class batches. We select 50 random classes from ImageNet and gather batches of 50 training samples from each class.

In Figure 8, shown in blue, NMF applied to single-class batches, has a denoising effect and improves over the baseline accuracy of the batch, shown as a dashed line. As the constraint on $k$ is relaxed, that accuracy drops back to its baseline level.

We contrast this behavior with the one shown in green when using multi-class batches. Here, only when $k$ is large do we regain baseline accuracy, and sensitivity to ablation is similarly diminished. This is due to the critical role non-linearity plays in separating the different classes. Ablating the NMF directions dramatically reduces classification accuracy. Finally, in Figure 8 (c) we show there is a significant per-class correlation (Pearson $r = 0.78$ ) between NMF AuC and accuracy on test accuracy on batches from the ImageNet test set.

## 6.4 EARLY STOPPING FOR FEW-SHOT LEARNING

In this experiment we use NMF-based early stopping to improve neural network accuracy in the context of few-shot learning, i.e., learning with very few samples per output label. We choose this setting since it is representative of data scarcity, where one would like to use all available data for training, rather than holding some out for validation.

We demonstrate this on the case of MNIST LeCun et al. (1998) digits with only 2 samples per class, which results in a training set of 20 samples. For early stopping with NMF, we set a very simple grid over $k$, a single point at $k = 1$. Thau NMf AuC thus simply becomes the training accuracy when compressed with NMF $k = 1$.

In Figure 9 it is evident that *training* set accuracy with NMF $k = 1$ shows similar gradients as the *test* set accuracy. Based on this observation, as before, we extract the first peak of the smoothed NMF curve, and stop there. Results are shown in Table 2 for 10 runs with randomly sampled training sets. We compare accuracy at our early stopping point to the best test set accuracy detected throughout the run (Best case), the average test set accuracy where the training set accuracy is 1 (Average case), and similarly the lowest test set accuracy where the set accuracy is 1 (Worst case).

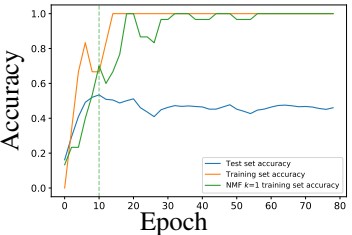 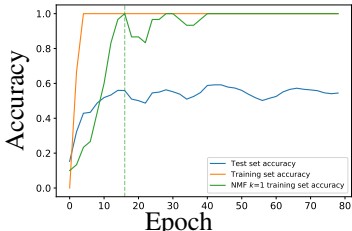

Figure 9: **NMF early stopping for few-shot learning** of MNIST digits with only 20 samples. By observing the *training* set accuracy under NMF $k = 1$ compression, we are able to correctly guess the gradient of the *test* set accuracy. We use this to perform early stopping with the simple heuristic of stopping at the first peak, which leads to improved accuracy as shown in Tabel 2.

| | | Our network | | | Kimura et al. (2018) |
|---|---|---|---|---|---|
| | | Early stop | Best case | Avg. case | Worst case | |
| Accuracy | | 54.8 | 57.3 | 50.9 | 29.4 | 53.9 |
| % of best case | mean | 0.954 | 1.000 | 0.887 | 0.515 | - |
| | std | 0.023 | 0.000 | 0.022 | 0.115 | - |

Table 2: **NMF early stopping for few-shot learning** of MNIST digits with only 20 samples. By observing the *training* set accuracy under NMF $k = 1$ compression, we are able to correctly guess the gradient of the *test* set accuracy. Early stopping at the first peak consistently improves accuracy.

In the first row of Table 2 we see that on average our method significantly improves over not using early stopping, and is on par with a recently proposed method specifically deigned for few-short learning. Furthermore, in the last two rows we show that per sampled training set, early stopping consistently improves accuracy.

## 6.5 NMF RECONSTRUCTION ERROR

In Figure 2 we show for every layer the area under the curve (AuC) of its $k$ vs. classification accuracy curve. However in addition to the accuracy, the NMF reconstruction itself is also a quantity of interest. There main difficulty involved with interpreting the NMF error is scale. The error depends on the magnitude of the activations, which varies across networks, layers and even channels.

In Figure 10 (a) and (b) we show the raw and normalized NMF reconstruction error, i.e. $\|\boldsymbol{A} - \tilde{\boldsymbol{A}}\|_2$ and $\frac{\|\boldsymbol{A} - \tilde{\boldsymbol{A}}\|_2}{\|\boldsymbol{A}\|_2}$ respectively. Measurements are taken over the same networks discussed in Figure 2. Observing the normalized values reveals that, proportionally, activation matrices become harder to

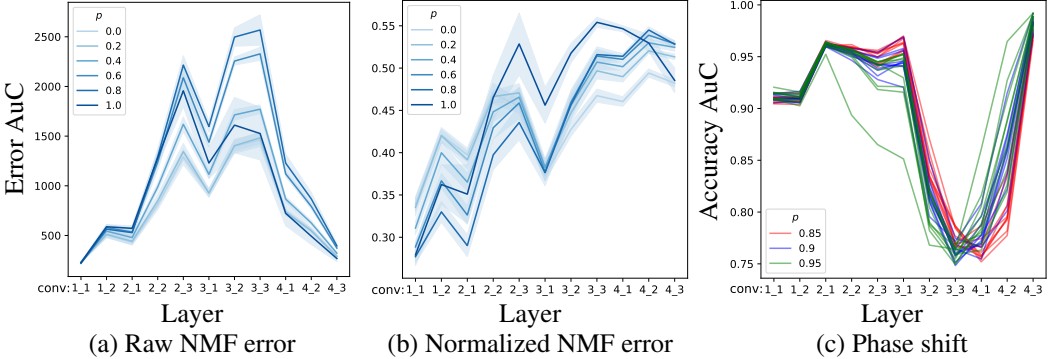

(a) Raw NMF error      (b) Normalized NMF error      (c) Phase shift

Figure 10: **NMF reconstruction error and extreme memorization.** Layer-by-layer view of (a) raw and (b) normalized NMF reconstruction errors, which NMF is trying to minimize. As we again notice the outlier behavior of networks trained with label randomization $p = 1$, in (c) we localize the transition between the two regimes to around $p = 0.9$.

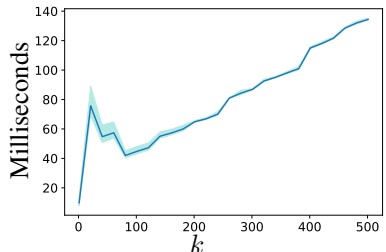

Figure 11: **NMF runtime on a typical ImageNet batch.** Thanks to GPU acceleration, NMF with multiplicative updates can be run to convergence in reasonable time.

approximate with depth, with an interesting interaction between the memorization level and depth. The error in absolute terms echo the accuracy curve, with $p = 1$ again presenting outlier behavior. Returning to accuracy measurement as in 2 (c), sampling $p$ more densely reveals in Figure 10 (c) that a phase shift occurs around $p = 0.9$, where networks "shift" their memorization to earlier layers.

## 6.6 NMF COMPUTATIONAL OVERHEAD

Applying NMF compression to large matrices naturally incurs certain overhead. We find, however, that our implementation of the multiplicative update algorithm Lee & Seung (2001) runs in reasonable time thanks to GPU acceleration.

We report timing results for typical batch used for VGG-19, i.e., 100 samples of 224×224 color images. At layer conv5_4 activations form a tensor of size $100 \times 14 \times 14$, which we flatten to a matrix of size $19600 \times 512$. In Figure 11 we show the timing curve for this batch as we increase $k$, using an NVIDIA Titan X card. As can be seen, at $k = 500$ processing of the batch to convergence requires 197 milliseconds on average.

For a batch of $32x32$ CIFAR-10 images, where the deep feature maps are, say, $8 \times 8$, the batch processing time drop to approximately 135 milliseconds for $k = 500$. Sweeping over *all* values of $k$ with an interval of 20 therefore takes about 2 seconds.

The final runtime depends on the number of classes sampled, and the granularity of the grid over $k$. We found our measurements to be robust to heavy subsampling of both.

