# OpenReview forum: "Detecting Memorization in ReLU Networks"
_ICLR.cc/2019/Conference_

### Official Review · AnonReviewer1 · 2018-11-01
**Review for "Detecting Memorization in ReLU Networks"**

**Rating:** 9
**Confidence:** 5

**Review:**

This paper aims to distinguish between networks which memorize and those with generalize by introducing a new detection method based on NMF. They evaluate this method across a number of datasets and provide comparisons to both PCA and random ablations (as in Morcos et al., 2018), finding that NMF outperforms both. Finally, they show that NMF is well-correlated with generalization error and can be used for early stopping.

This is an overall excellent paper. The writing is clear and and focused, and the experiments are careful and rigorous. The discussion of prior work is thorough. The question of how to detect memorization in DNNs is one of great interest, and this makes nice steps towards this goal. As such, it will likely have significant impact.

Major comments:

1) The early stopping section could benefit from more experiments. In particular, it would be helpful to see a scatter plot of the time of peak test loss as a function of NMF/Ablation AuC local maxima and to measure the correlation between these rather than simply showing 3 examples.

Minor comments:

1) While the comparisons to random ablations are mostly fair, it is worth noting that the variance on random ablations appears to be lower than that of NMF and PCA.

2) The error bars on the plots are often hard to see. Increasing the transparency somewhat would be helpful.

Typos:

1) Section 1, third paragraph: “We show that networks that networks that generalize…” should be “We show that networks that generalize...”

2) Section 3.1, third paragraph: “Because threshold is the…” should be “Because thresholding is the…”

3) Section 3.2, third paragraph: “In the most non-linear case we would…” should be “In the most non-linear case, we would…”

4) Figure 2 caption: “...with increasing level of…” should be “...with increasing levels of…”

5) Section 4.1.1, second to last line of last paragraph: missing space before final sentence

6) Figure 4a label: “Fahsion-MNIST” should be “Fashion-MNIST”

---

> ### Author Response · Authors · 2018-11-21
> **Response to reviewer 1**
>
> We thank the reviewer for the helpful review!
>
> "The early stopping section could benefit from more experiments. In particular, it would be helpful to see a scatter plot of the time of peak test loss as a function of NMF/Ablation AuC local maxima and to measure the correlation between these rather than simply showing 3 examples."
> As suggested, we have performed more experiments for early stopping, and summarize the results in Figure 6c. We have also added early stopping experiments in the setting of few-shot learning in section 6.4 of the appendix (specifically Figure 9 and Table 2).
>
> "it is worth noting that the variance on random ablations appears to be lower than that of NMF and PCA."
> Indeed, we added this comment to the text, as well as a note on computation time.
>
> "The error bars on the plots are often hard to see"
> Thank you for pointing this out. We have improved the visibility of error bars in all figures.

---

### Official Review · AnonReviewer2 · 2018-11-02
**bad clustering == memorization?**

**Rating:** 6
**Confidence:** 4

**Review:**

This paper propose a new way of analyzing the robustness of neural network layers by measuring the level of "non-linearity" in the activation patterns on samples belonging to the same class, and correlate that to the level of "memorization" and generalization.

More specifically, the paper argues that a good representation cluster all the samples in a class together, therefore, in higher layers, the activation pattern of samples from the same class will be almost identical. In this case, the activation matrix will have a small non-negative rank. An approximation algorithm (via non-negative matrix factorization) is then used to compute the robustness and evaluate the robustness (by replacing the activation matrix with its low rank non-negative activation) is measured in a number of experiments with different amount of random label corruptions. The experiments show that networks trained on random labels are less robust than networks trained on true labels.

While the concept is interesting, I find the arguments in the paper a bit vague, and the usefulness of the algorithm might be hampered by its computation complexity, which is not discussed in the paper.

First of all, the paper lacks a clear notion of "memorization". While it is generally accepted that learns on random labels can be called "memorization", the paper seem to be defining it as how well is the network clustering points from the same class. Several questions need to be addressed in order for this notion to be justified:

1. Are (well generalized) networks really clustering samples of the same class to a centroid? It would be great if some empirical verifications are shown. Because the networks are using linear classifier in the last layer to classify the samples, it seems only linearly separability would be suffice for the work, which does not necessarily imply clustering.

2. Given two networks (of the same architecture), assume somehow network-1 decides to use the first 9 layers to compute a well clustered representation, while network-2 decides to use the first 5 layers to do the same thing. Do we say network-1 is (more) memorizing in this case?

3. The notion seems to be more about the underlying task than about the networks. Given the measurement, if a task is more complicated, meaning the input samples in the class have higher variance and requiring more efforts to cluster, then it seems the network will be doing more memorization. In other words, while networks will be doing more memorization when comparing a random label task to a true label task, it might also be "doing more memorization" when comparing learning on imagenet to learning on MNIST / CIFAR. One the one hand, this does not seem to fit our "intuition" about memorization; on the other hand, the heavy dependency on the underlying data distribution makes it difficult to compare results learned on different data -- especially since the measurements are based on per-class samples, "random labels" and "true labels" have very different class-conditional distributions.

I also have some questions about Figure 2(c). I will continue numbering the question for easier discussion.

4. Why for all cases, the lower layers all have higher AUC than the higher layers (except the last one)? The argument given in the paper is that the lower layers are the feature extraction phase while the upper layers are memorization phase. I think if clearly verified, this is a very interesting observation. But the paper currently do not have experiments to verify the hypothesis. Also more studies on this with different networks would be good. For example, with deeper networks, does the feature extraction phase include more layers?

5. The p=1 and p<1 curves seem to be very different. If one is to sample more densely between p=0.8 and p=1, would there still be a clear phase transition?

Some other questions:

6. Can you add discussions to the computation requirements for the proposed analysis? This is especially important for the cases where the analysis is used during training as tools to help deciding early stopping.

7. For the early stopping experiment, the main text says "These include the test error (in blue)" while in the figure the label axis is "Test loss". I'm assuming it is the cross entropy loss given the values are greater than 1. In this case, can you show in parallel the same plots in error rate, as the test error is more important than the test loss and the test loss could sometimes be artificially huge due to high confident mistakes on ambiguous test examples.

Some minor issues:

* Please proof read the paper for typos. E.g. on the 3rd paragraph of the 1st page: "that networks that networks that".

* The convention with subplots seem to be putting sub-captions under the figures, not above.

---

> ### Author Response · Authors · 2018-11-21
> **Response to reviewer 2 (1/2)**
>
> We thank the reviewer for the insightful comments!
>
> "the paper lacks a clear notion of "memorization"... The paper seem to be defining it as how well is the network clustering points from the same class"
> We have clarified in the text what we (Introduction, second-to-last paragraph). In particular, our definition of memorization is the network implicitly learning a specific mapping from the sample with index i to the class with index j - a "rule" which does not benefit the network in terms of improving its generalization.
> We then suggest that good clustering within approximately linear regions of deep feature space correlates with absence of memorization.
>
> "...the paper argues that a good representation cluster all the samples in a class together..."
> "Are (well generalized) networks really clustering samples of the same class to a centroid?"
> NMF basis vectors are not centroids in the k-means sense. In particular, a datapoint can be associated with multiple NMF basis vectors and at various scales. Furthermore, we do not claim the existence of a single class-specific cluster, but rather a distribution into a small number of clusters (approximated by k).
>
> "Because the networks are using linear classifier in the last layer to classify the samples, it seems only linearly separability would be suffice for the work, which does not necessarily imply clustering."
> Linear separability of last-layer activations is indeed all that is required for correct classification. While all the networks we studied achieve perfect linear separability on their *training* data, this is clearly not the case for their validation and test data. Training set linear-separability is therefore not a sufficient condition for test set linear-separability. Our aim is to find additional properties of neural network feature space that are indicative of last-layer linear-separability of test data.
> There are many proposals for what makes a "good" feature space in that regard [1,2,3]. In this paper we propose, and give supporting empirical evidence, that feature spaces characterized by a low non-negative rank of single-class activation matrices memorize less and generalize more.
>
> "Given two networks (of the same architecture), assume somehow network-1 decides to use the first 9 layers to compute a well clustered representation, while network-2 decides to use the first 5 layers to do the same thing. Do we say network-1 is (more) memorizing in this case?"
> In such a situation we would prefer the network that more quickly reduces the non-negative rank, because it is a simpler model of the data. This view is based on the general principle of Occam's razor, and is made concrete with our method.
>
> "the notion seems to be more about the underlying task... if a task is more complicated, meaning the input samples in the class have higher variance and requiring more efforts to cluster, then it seems the network will be doing more memorization... when comparing learning on imagenet to learning on MNIST / CIFAR"
> It is absolutely true that some datasets are more complicated than others. We therefore do not propose a global measure for memorization, but rather a comparative measure to evaluate competing networks of the same architecture on the same dataset.
>
> "Why for all cases, the lower layers all have higher AUC than the higher layers (except the last one)?"
> In Figure 2.c, note that for the case p=0, most lower layers actually have lower AuC than higher layers. We interpret this as meaning that without artificially inducing memorization, the network better clusters activations from layer to layer.
>
> "The argument given in the paper is that the lower layers are the feature extraction phase while the upper layers are memorization phase."
> Our hypothesis is based on the statistical similarity of early layers in our compression studies. We altered the text to emphasize it is indeed currently a hypothesis.

---

> > ### Comment · AnonReviewer2 · 2018-11-26
> > **Thanks for the rebuttal**
> >
> > Thanks the author for the rebuttal. It clarifies most of my concerns. I have updated my score. Though I still think this paper is around borderline as the method can only be used to comparatively study the memorization effects on the same dataset and for the same network architecture, which could already be achieved via simpler previous methods like PCA, ablation studies or even measuring the norms of the layer weights.
> >
> > If the paper is accepted, I would like to request the author to modify the paper title to be more specific. The current title sounds like the grand challenge of detecting memorization has been solved in this paper, which is a bit misleading as the general case is still quite open. Maybe the title could be made more specific by mentioning this is an approach based on NMF, or maybe indicating that it only works as a relative comparison on identical tasks and network architectures.

---

> > > ### Author Response · Authors · 2018-11-29
> > > **Response to Reviewer2**
> > >
> > > Thank you again for the helpful comments.
> > >
> > > We agree that the task of detecting memorization in general is not conclusively resolved by our paper.
> > > The measurements based on parameter-norms, mentioned in the paper, have very limited usefulness in practical applications. We also note that the PCA method was proposed in our work. This was a means to provide a baseline for comparison. However, we can change the title to "On detecting..." if there is consensus among the reviewers.

---

> ### Author Response · Authors · 2018-11-21
> **Response to reviewer 2 (2/2)**
>
> "The p=1 and p<1 curves seem to be very different. If one is to sample more densely between p=0.8 and p=1, would there still be a clear phase transition?"
> This is indeed interesting. We have added plots regarding the phase shift between the cases p=0.8 and p=1 in section 6.5 of the appendix. In the case of CIFAR-10 and our network architecture, this change happens around p=0.9.
>
> "Can you add discussions to the computation requirements for the proposed analysis?"
> As suggested, we have added a section to the appendix (6.6) discussing the computational complexity of our algorithm. While NMF naturally incurs certain overhead, our implementation runs in reasonable time thanks to GPU acceleration.
>
> "For the early stopping experiment, the main text says "These include the test error (in blue)" while in the figure the label axis is "Test loss""
> Good catch, we fixed the text.
>
> "...can you show in parallel the same plots in error rate, as the test error is more important than the test loss"
> As suggested, we show the usefulness of our method to early stopping by comparing accuracy curves in newly added section 6.4 of the appendix.
>
> "The convention with subplots seem to be putting sub-captions under the figures, not above"
> It is now fixed.
>
> [1] Bengio et al. "Representation learning: A review and new perspectives." 2013
> [2] Tishby et al. "Deep learning and the information bottleneck principle." 2015
> [3] Amjad, et al. "How (Not) To Train Your Neural Network Using the Information Bottleneck Principle." 2018

---

### Official Review · AnonReviewer3 · 2018-11-03
**Very interesting but not yet a complete work**

**Rating:** 5
**Confidence:** 4

**Review:**

The contribution of the paper is in proposing a quantitative measure of memorization based on the assumption that the activations at the deeper layers of a *generalizing* deep network should be invariant to intra-class variations. The measure corresponds to how well can the activation matrix of a batch be approximated by a low-rank decomposition. The paper proposes to use approximate non-negative matrix factorization and compares it to PCA. As for “wellness” it uses the final accuracy of the network after the activation is approximated in some layer(s).

The composition of the paper and its writing makes it an easy read. The work is novel in the way it proposes to measure memorization to the best of the reviewer’s knowledge. However, the novel insights and/or the practical usefulness of the proposed method seem very limited. Also, there are many questions that comes to my mind that I would appreciate the authors to address:

Specific questions:
The experimental setup is unclear:
Is the linearization-batch taken from the training set or the test set?
If it is taken from the training set, for the case that p>0 (noisy labels), is the batch of a single class obtained from noisy labels or non-noisy labels?
For the experiments, is there only one fixed batch used? How is this batch selected? How sensitive the evaluation is to the selection of the batch members and its size?
Do the batches cover the whole set?

- Figure 2.a and 2.b: How come all networks with different label noise levels end up with the same (100%?) accuracy? Are the fixed samples different for each p? (class labels change for each p).

- Figure 2.c: Why should the performance drop more when linearizing the middle layers (3_2:4_2) than the earlier layers. This seems to be in violation of the assumption about class invariance in deeper layers.

- When k=1 for NMF and PCA, the difference of the activations for different samples becomes a matter of scale. In this case, shouldn’t all classifications become the same for all samples? How does this affect the accuracy? Does it make the evaluation very sensitive to the sampling of the batch? It would be interesting to study the property of the basis obtained in this border case. The same questions can be studied as one gradually increases k.
Section 4.2: It starts with the sentence “In this section we show our technique is useful for predicting good generalization in a more realistic setting“. Indeed, the high correlation of the test performance and the proposed memorization measure in this section is very interesting. However, as for usefulness, it does not seem to provide a better criterion for early stopping or other practicalities of ReLU networks.

- An experiment describing how well are the approximations (i.e. activation matrix reconstruction error) and how that correlate with memorization is missing.

Some general questions that come to my mind:

- the paper assumes (e.g. in page 4) that “When single-class batches are not approximately linear, even in deep layers, we take this as evidence of memorization”. I have a concern here. Apart from the last layer, this form of simplicity of the support for the intermediate layers of a good classifier does not seem to be *necessarily*. That is, it seems to me that as long as the activation matrix of each class is linearly separable from the activation matrices of the others, there is no need for it to become simpler (by reducing the intra-class variations at the deep layers) for the classification loss to be minimized. Does this mean the paper’s assumption for memorization is not necessarily valid?

- The paper relates the memorization to the extent of local linearity of a deep ReLU network. ReLU networks represent piece-wise linear functions. Thus, in order for this relation to be drawn, probably different linear regions (polytopes in the input space) should be considered for the linearization of the activation matrix. In that regard, how can this empirical measurement be translated into a more formal linearity of the global function?

- The rc number as well as the rank k of a good approximation directly depend on the number of samples in the batch. How can one obtain a measure that is independent of the number of samples in the batch?


Summary judgment:

The paper puts forward an interesting observation using a novel approach. However there are questions about the experiments, discussions around the experiments and the usefulness of the observation for training better models and/or giving additional insights to what we know. Considering that, I think the paper would make a very good workshop paper but needs more work to address the bar of an ICLR conference paper. But I am open to discussion with the authors and other reviewers.

---

> ### Author Response · Authors · 2018-11-21
> **Response to Reviewer 3 (1/2)**
>
> We thank the reviewer for the constructive feedback!
>
> "The experimental setup is unclear"
> We have cleared up the ambiguities regarding our experimental setup (section 4.1, paragraph 3), noting that:
> - We use training-set batches
> - Single-class batches are sampled w.r.t. to the training label, i.e., the random/noisy labels for p>0
> - For every value of p we produced a fixed set of random labels.
> - We do not use a fixed batch. We randomly sample batches (up to the class label) for each net.
> - Batches are not exhaustive over the dataset.
> - In our experiments with various batch sizes (20-100) we did not notice significant difference. We used a batch size of 50 through out the paper.
>
> "How come all networks with different label noise levels end up with the same (100%?) accuracy?"
> The y-axis in Figure 2 is *training* set accuracy. All network manage to (over)fit their training data regardless of the level of label randomization [4], and therefore show perfect accuracy under weak/no compression.
>
> "Why should the performance drop more when linearizing the middle layers (3_2:4_2) than the earlier layers."
> This is because we perform factorization across the channel dimension of the CNN. In early layers most of the information is spread across the spatial dimensions, but as the effective receptive field grows and the spatial resolution of the feature maps decreases with depth, the channel dimension holds more and more information.
>
> "When k=1 for NMF and PCA, the difference of the activations for different samples becomes a matter of scale. ... shouldn’t all classifications become the same for all samples?"
> While with k=1 all activations do point in the same direction in feature space, classification is ultimately performed over a whole feature map, where the spatial arrangement and different scales of activations leads to different predictions.
>
> "It would be interesting to study the property of the basis obtained in this border case. The same questions can be studied as one gradually increases k."
> Qualitative examination of the NMF basis with small values of k is has been undertaken in [5] where in deep layers basis directions are shown to correspond with semantic concepts.
>
> "it does not seem to provide a better criterion for early stopping"
> Validation error is the gold standard for stopping criteria, and so when a validation set is available, we do not propose a better alternative. By showing good correlation with validation error, our method is useful where a validation set is not available. To demonstrate this point we added section 6.4 to the appendix, where we perform few-shot learning on MNIST with only 20 images for training. Early stopping with NMF consistently improves the final accuracy of the network.
>
> "An experiment describing how well are the approximations and how that correlate with memorization is missing"
> We show and discuss NMF reconstruction error plots in the appendix (section 6.5). The main difficulty in interpreting the NMF error is scale. The error depends on the magnitude of the activations, which varies across networks, layers and even channels. Network accuracy, on the other hand, is more interpretable and is ultimately the quantity of interest w.r.t. the effect on the neural network.

---

> > ### Comment · AnonReviewer3 · 2018-11-27
> > **Thanks for the additional experiments and comments**
> >
> > I appreciate the authors effort in addressing the raised issues by revising the paper, doing additional experiments and replying to comments. I believe the paper has improved, however I see the following main issues are still outstanding:
> >
> > 1)  “Batches are not exhaustive over the dataset.”: 1.a) how sensitive is the new training set accuracy to the choice of the batches 1.b) how many batches do you consider per experiment? 1.c) how sensitive is the study to the number of batches?
> >
> > 2) “This is because we perform factorization across the channel dimension of the CNN. In early layers most of the information is spread across the spatial dimensions, but as the effective receptive field grows and the spatial resolution of the feature maps decreases with depth, the channel dimension holds more and more information.” This is not a satisfactory explanation unless additional experiments are provided. Figure 2.c shows a rapid drop from conv_3_1 to conv_3_2 and then a sharp increase from 4_1 to 4_3. It sounds ad-hoc to me to assume the first drop is a sudden turn of spatial class-information to non-spatial (channel-wise) class-information and the increase is due to just being close to the final classification layer. While this conjecture can be true, proper experiments should be conducted to confirm this.
> >
> > 3) “While with k=1 all activations do point in the same direction in feature space, classification is ultimately performed over a whole feature map, where the spatial arrangement and different scales of activations leads to different predictions.” what is the difference between a “whole feature map” and “activations”? Do you mean that for the convolutional layers the approximation is done separately per channel and thus there are different scaling factors per channel for each sample?
> >
> > 4) “our method is useful where a validation set is not available. [...] few-shot learning on MNIST with only 20 images ” I appreciate the additional experiments provided in Ap. 6.4 However, for this to be true, it is important to show that the hyperparameters used (smoothing factor for the approximated training set accuracy plot, batchsize, k, and maybe others) are independent of the dataset. Otherwise, it would diminish the benefit of not needing a validation set since one needs to find the best values on a heldout set.

---

> > > ### Author Response · Authors · 2018-11-29
> > > **Response to Reviewer3**
> > >
> > > Thank you again for the helpful comments.
> > >
> > > 1) "1.a) how sensitive is the new training set accuracy to the choice of the batches 1.b) how many batches do you consider per experiment? 1.c) how sensitive is the study to the number of batches?"
> > >
> > > In our experiments, we used one batch per class, per randomization probability p, and per network instance.
> > > In datasets with 10 outputs (Fashion-MNIST, CIFAR10, SVHN, UrbanSounds) we set the number of batches per network to 10, i.e. 10 multi-class batches or 10 single-class batches (1 batch per class). This was to equate the amount of data seen by our method.
> > > Additionally, the batches were allowed to vary across the 10 network instances tested and also per label randomization probability p.
> > > This means that to evaluate each accuracy tradeoff in out experiments (Figures 2, 3, 4, 7 and 10) we randomly sampled 100 batches.
> > > We note that each batch contained 50 samples throughout our experiments.
> > >
> > > This choice had no significant impact on our results because of the very small variance across batches.
> > > For instance, we measured the AuC of the curves in Figure 3(a) over 6 different runs, i.e., each time using a different per-class batch. Below we report, per label randomization probability p, the mean and the standard deviation:
> > >
> > >           	 mean	  std
> > > p=0.0	0.9756	0.0008
> > > p=0.2	0.8779	0.0013
> > > p=0.4	0.7704	0.0039
> > > p=0.6	0.6847	0.0066
> > > p=0.8	0.6381	0.0038
> > >
> > > We understand the importance of clarity in our experiments, and we will discuss the batch sensitivity and accuracy variance in the paper.
> > >
> > > 2) “This is not a satisfactory explanation unless additional experiments are provided. Figure 2.c shows a rapid drop from conv_3_1 to conv_3_2 and then a sharp increase from 4_1 to 4_3. It sounds ad-hoc to me to assume the first drop is a sudden turn of spatial class-information to non-spatial (channel-wise) class-information and the increase is due to just being close to the final classification layer. While this conjecture can be true, proper experiments should be conducted to confirm this."
> > >
> > > As suggested by the reviewer, to support our hypothesis, we performed an extensive layer-by-layer evaluation on the fully-connected network we have trained on Fashion-MNIST.
> > >
> > > Except for the network architecture and dataset, the rest of the experiment is identical to the experiment in Figure 2. This is to validate the effect of spatial arrangement in the early layers of our convolutional network.
> > >
> > > The results (seen here: https://ibb.co/K6GQbRP ) show that in this case the early layers are less robust to compression, as per the reviewer's initial intuition, which supports our hypothesis as to the phenomenon observed in Figure 2.
> > >
> > >
> > > 3) "what is the difference between a “whole feature map” and “activations”? Do you mean that for the convolutional layers the approximation is done separately per channel and thus there are different scaling factors per channel for each sample?"
> > >
> > > We apologize for the ambiguity in our terminology. By 'activations' we were referring to individual C-dimensional vectors comprising the NxCxHxW activation tensor.
> > > For NMF and PCA we consider every C-dimensional vector as a single datapoint (of which there are N*H*W), while for classification the network views every 1xCxHxW block as a single datapoint. So while every C-dimensional vector is pointing in the same direction, there is variability due to scale and spatial arrangement within each 1xCxHxW block.
> > >
> > >
> > > 4) “I appreciate the additional experiments provided in Ap. 6.4 However, for this to be true, it is important to show that the hyperparameters used (smoothing factor for the approximated training set accuracy plot, batchsize, k, and maybe others) are independent of the dataset. Otherwise, it would diminish the benefit of not needing a validation set since one needs to find the best values on a heldout set."
> > >
> > > We agree with the reviewer that more investigation and experiments are required for the practical implementation of the early stopping application of our approach. We will clarify this in our paper.

---

> > > > ### Comment · AnonReviewer3 · 2018-12-02
> > > > **no title**
> > > >
> > > > I comment using the numbering we have in the thread above:
> > > >
> > > > 1) I find the explanation and the additional experiment convincing.
> > > >
> > > > 2) I find the new experiment interesting and *partially* supporting the conjecture about the phases.
> > > >
> > > > 3) Now I understand and this explains the behavior. However, the text in the method section calls $A$ a "layer activation matrix". That is each row represent a sample and the columns the units in that layer. Next, $A$ is what that gets factorized. This does not imply the interpretation in the previous comment about each spatial location becoming a separate row in the $A$ matrix and, thus, is an incorrect representation of the method (for convolutional layers). So, the method section needs to be rewritten to make this clear.
> > > >
> > > > 4) This point is important since the usefulness of the proposed early stopping procedure is the main empirical contribution of the submission.
> > > >
> > > > Final take: I deeply appreciate the authors effort in addressing these issues. Including all these comments and experiments in the main manuscript will make the work more clear and convincing. However, unfortunately, I still think the work's usefulness from both conceptual and empirical aspects is lacking. From the conceptual point of view, it does not put forward a systematic *and* novel perspective, the phase study of layers is novel/interesting but not systematic, the linearity analysis using NMF is systematic in general but does not put forward novel findings. The empirical usefulness of the proposed method is also at question (see point 4 above). Also, the manuscript had many unclear points, many (or all) of which are cleared through the discussion but requires a reorganization for them to be properly integrated in the text and potentially find other unclear parts.
> > > >
> > > > So, all in all, given my final view above, I would prefer to see the paper get accepted when all these points are properly addressed (which should be possible for the next ML conference) but would not be too disappointed if it gets accepted to this ICLR conference track.

---

> ### Author Response · Authors · 2018-11-21
> **Response to Reviewer 3 (2/2)**
>
> "Apart from the last layer, this form of simplicity of the support for the intermediate layers of a good classifier does not seem to be *necessarily*.  ...as long as the activation matrix of each class is linearly separable ... there is no need for it to become simpler"
> Linear separability of last-layer activations is indeed all that is required for correct classification. While all the networks we studied achieve perfect linear separability on their *training* data, this is clearly not the case for their validation and test data. Training set linear-separability is therefore not a sufficient condition for test set linear-separability. Our aim is to find additional properties of neural network feature space that are indicative of last-layer linear-separability of test data.
> There are many proposals for what makes a "good" feature space in that regard [1,2,3]. In this paper we propose, and give supporting empirical evidence, that feature spaces characterized by a low non-negative rank of single-class activation matrices memorize less and generalize more.
>
> "different linear regions (polytopes in the input space) should be considered for the linearization of the activation matrix... how can this empirical measurement be translated into a more formal linearity of the global function?"
> Under this geometric lens, our observation is that for a particular point cloud, i.e., one associated with a single-class, there is a small number of deep local polytopes where most of the data "fits" without being non-linearly projected into the polytope by ReLU. However, associating these deep polytopes with polytopes in input space is a non-trivial problem which is concerned with "network interpretability", an active area of research.
>
> "How can one obtain a measure that is independent of the number of samples in the batch?"
> The approach presented in the paper is based on the properties of activation matrices, which inherently depends on a batch and therefore its size. However as mentioned, we have found our measurements to be robust across wide range of batch sizes .
>
> "there are questions about ... the usefulness of the observation for training better models and/or giving additional insights to what we know"
> We show that memorization and generalization are correlated with non-negative rank of activation matrices. A next step would be to regularize this or a dependent quantity. While, the non-negative rank and rectangle cover number are NP-hard to compute, we believe practical regularizers could be derived from this connection. This is a direction of future work that we intend to pursue, and we see value in sharing these results with the wider community to spark further interest in this direction.
>
> [1] Bengio et al. "Representation learning: A review and new perspectives."  2013
> [2] Tishby et al. "Deep learning and the information bottleneck principle."  2015
> [3] Amjad, et al. "How (Not) To Train Your Neural Network Using the Information Bottleneck Principle." 2018
> [4] Zhang et al. "Understanding deep learning requires rethinking generalization." 2016
> [5] Collins et al. "Deep feature factorization for concept discovery." 2018.

---

### Author Response · Authors · 2018-11-21
**General comment**

We would like to thank all reviewers for their thorough and insightful feedback. We are glad that the reviewers found our approach "novel" and "very interesting", and the paper  "clear and focused".

We have made revisions and additions to the paper guided by the reviews. In particular, we have added more experiments for early stopping (also in the few-shot learning setting), added plots for the NMF approximation error w.r.t. memorization, and discuss the computational cost involved with our method.

Below we discuss each of the reviewers' comments in detail.

---

### Meta-Review · Area_Chair1 · 2018-12-13

**Confidence:** 2
**Recommendation:** Reject

**Metareview:**

This paper proposes a new measure to detect memorization based on how well the activations of the network are approximated by a low-rank decomposition. They compare decompositions and find that non-negative matrix factorization provides the best results. They evaluate of several datasets and show that the measure is well correlated with generalization and can be used for early stopping. All reviewers found the work novel, but there were concerns about the usefulness of the method, the experimental setup and the assumptions made. Some of these concerns were addressed by the revisions but concerns about usefulness and insights remained. These issues need to be properly addressed before acceptance.